# A causal role for right temporo-parietal junction in signaling moral conflict

**Ignacio Obeso[1,2†], Marius Moisa[3], Christian C Ruff[3], Jean-Claude Dreher[1]***

[1]Neuroeconomics, reward and decision making group, Institut des Sciences Cognitives Marc Jeannerod, CNRS, UMR 5229 and Université Claude Bernard (UCBL), Lyon 1, Bron, France; [2]Fundación de Investigación HM Hospitales, HM Hospitales – Centro Integral en Neurociencias HM CINAC, Madrid, Spain; [3]Laboratory for Social and Neural Systems Research, Department of Economics, University of Zurich, Zurich, Switzerland

**Abstract** The right temporo-parietal junction (rTPJ) has been proposed to play a key role in guiding human altruistic behavior, but its precise functional contribution to altruism remains unclear. We aimed to disentangle three possible functions of the rTPJ for human altruism, namely: implementing the motivation to help, signaling conflicts between moral and material values, or representing social reputation concerns. Our novel donation-decision task consisted of decisions requiring trade-offs of either positive moral values and monetary cost when donating to a good cause, or negative moral values and monetary benefits when sending money to a bad cause. Disrupting the rTPJ using transcranial magnetic stimulation did not change the general motivation to give or to react to social reputation cues, but specifically reduced the behavioral impact of moral-material conflicts. These findings reveal that signaling moral-material conflict is a core rTPJ mechanism that may contribute to a variety of human moral behaviors.
DOI: https://doi.org/10.7554/eLife.40671.001

***For correspondence:**
dreher@isc.cnrs.fr

**Present address:** [†]Centro Integral en Neurociencias AC, Puerta del SurHM Hospitales and CEU-San Pablo University, Madrid, Spain

**Competing interests:** The authors declare that no competing interests exist.

## Introduction

Humans have the ability to forego material interests in the service of abstract moral values. This is evident in different aspects of human morality and social codes, such as fairness (*Knoch et al., 2006*), honesty (*Shalvi et al., 2012*) or altruism (*Fehr and Fischbacher, 2003*), which are widely accepted codes of conduct. Altruistic giving is an important feature of human social behavior since it is necessary for the formation of stable groups (*Fletcher and Zwick, 2004*; *Zisis et al., 2015*); this behavior nevertheless remains puzzling because giving away to others often directly opposes personal self-interest, which leads people to help others only in some circumstances determined by a precise context (*DePaulo et al., 1996*; *Shalvi et al., 2013*).

Evolutionary theories of social cognition propose that altruistic behavior may reflect the workings of dedicated neural networks that have evolved to facilitate cooperation in social contexts (*Boyd et al., 2003*; *Rilling et al., 2002*). Neuroimaging studies have shown that altruistic behavior leads to activity in a core brain network that includes portions of the prefrontal cortex (*Hare et al., 2010*; *Moll et al., 2006*; *van der Meulen et al., 2016*; *Waytz et al., 2012*) and the right temporo-parietal junction (rTPJ) (*Hare et al., 2010*; *Hutcherson et al., 2015*; *Jeurissen et al., 2014*; *Morishima et al., 2012*; *van der Meulen et al., 2016*). While there is consensus that all these areas play important roles for altruistic decisions (*Izuma, 2012*; *Schurz et al., 2014*), it is less clear what precise functional contributions each of these areas make and whether or not each area is causally necessary for altruistic behavior to occur.

In the present study, we focus on the rTPJ, a brain area thought to contribute to several of the cognitive operations underlying altruistic behaviour (*Izuma, 2012*; *Schurz et al., 2014*;

*Strombach et al., 2015*). We aim to test three different accounts of the rTPJ role in this respect. According to these accounts, the rTPJ may either be necessary for bringing to bear the motivation to do good to others ('other-regarding motivation', *Hare et al., 2010*; *Hutcherson et al., 2015*; *Jeurissen et al., 2014*; *Park et al., 2017*; *van der Meulen et al., 2016*), for representing the conflict between moral values and material concerns associated with sharing resources ('moral conflict', *Berns et al., 2012*; *Morishima et al., 2012*), or for managing one's social reputation by displaying socially desired behavior ('reputation', *Izuma, 2012*; *Izuma, 2013*; *Yomogida et al., 2017*).

To test the predictions of these three accounts, we employed a novel paradigm (*Figure 1*) requiring participants to accept or reject monetary transfers to two organizations (supporting morally 'good' or 'bad' causes) that were coupled with changes in the participants' financial payoff. One of these organizations was a charity perceived as having a high moral value that should be supported. Monetary transfers to this organization were coupled with a variable monetary cost to the participants (deducted from their initial endowment; see *Figure 1* and materials and methods). Choices

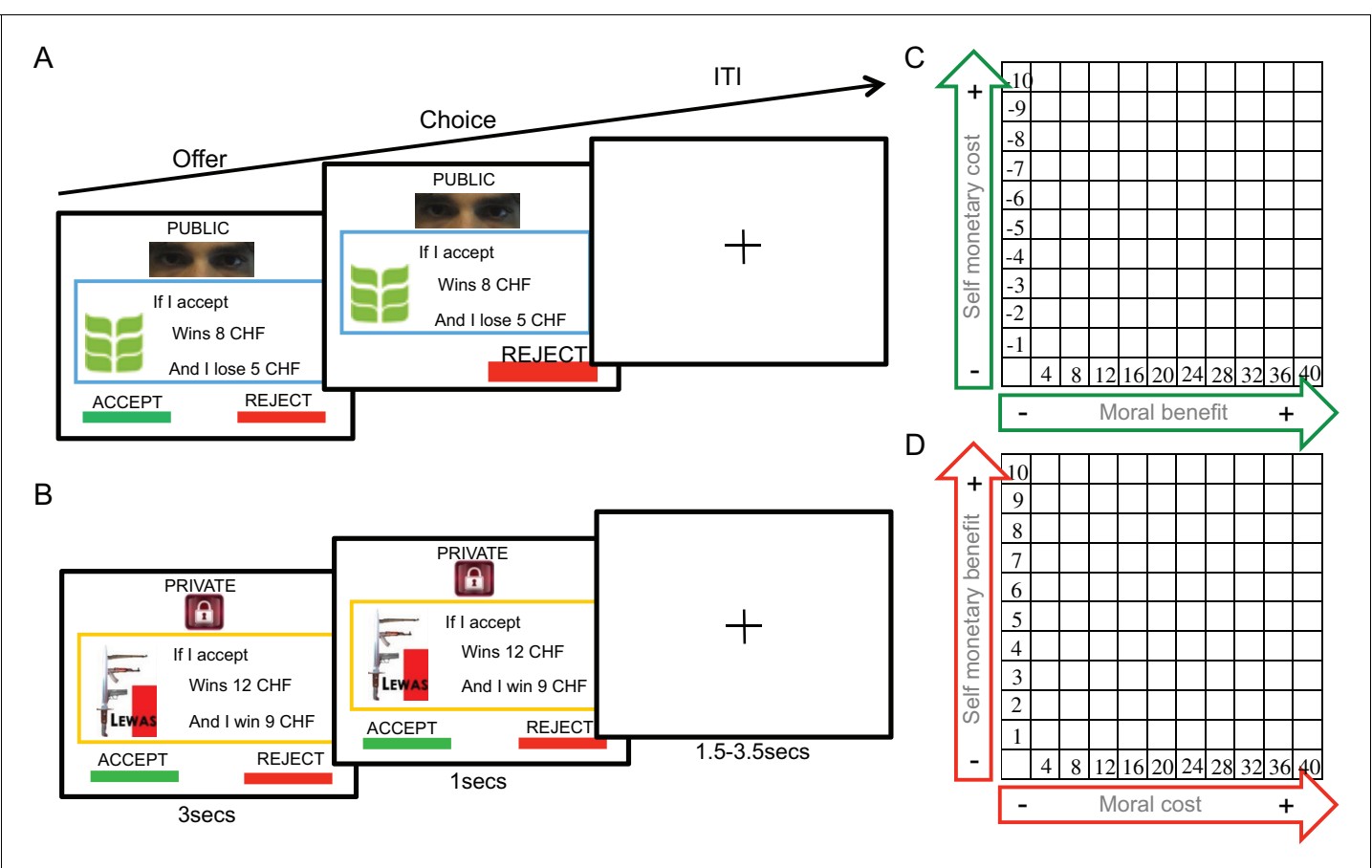

**Figure 1.** Donation task. (A) Presentation of moral context and public trial conditions, with green leaf logo from 'Bread for all' organization; (B) presentation of immoral context and private trial conditions, showing the 'Lewas' organization logo. (C) It shows the trial distribution of the moral context and D the trial distribution of the immoral context.

DOI: https://doi.org/10.7554/eLife.40671.002

The following figure supplements are available for figure 1:

**Figure supplement 1.** Subjects value rating of organization types used in the donation task.
DOI: https://doi.org/10.7554/eLife.40671.003

**Figure supplement 2.** Audience effects are not changed by TMS.
DOI: https://doi.org/10.7554/eLife.40671.004

**Figure supplement 3.** Reaction times for the different conditions show no significant differences [all F's > 1], suggesting that TMS did not lead to task disengagement or distraction.
DOI: https://doi.org/10.7554/eLife.40671.005

about transfers to this organization therefore required a comparison of the moral benefit of the transfer with the conflicting monetary cost. The other organization had a negative moral value that participants did not agree with, but transfers to this organization were coupled with a monetary payment to the subject. That is, participants had to weigh the moral cost of the transfer to the bad organization *versus* the conflicting personal financial benefit resulting from it. Additionally, both types of decisions were coupled with an audience manipulation to facilitate an 'audience effect': on half of the trials, an observer could see choices made by the participant, whereas choices were fully private on the other half of the trials. Concerns about the effects of the donation on social reputation should therefore be mainly prominent in the condition where choices were being watched by another person. This paradigm allowed us to compare opposing predictions of the three accounts given above, based on a rich literature proposing these different functions for the rTPJ in human altruistic behavior.

As for the first account, many recent findings are consistent with the interpretation that the rTPJ underlies other-regarding motivation during choices to do good to others. For instances, several studies have reported TPJ activation during choices to help others even though this is personally costly (*Hare et al., 2010*; *Hutcherson et al., 2015*); moreover, TPJ was significantly activated when participants chose to help others by showing social inclusion (*van der Meulen et al., 2016*) or by refraining from punishment (*Will et al., 2015*). Several of these reports show that the larger the TPJ activation (experimentally manipulated by different characteristics of the other person, that is, social distance), the more willing participants are to share money with the other person (*Hare et al., 2010*; *Hutcherson et al., 2015*; *Strombach et al., 2015*). Overall, these findings are consistent with the idea that TPJ activity underlies the motivation for other-regarding choices to help. This account would therefore predict that TMS should bias participants to transfer less money to either 'good' or 'bad' recipients, reflecting a reduced general motivation to help.

The second account relates to the potential role of the rTPJ in signaling conflicts between moral and material interests during helping decisions. For instance, the rTPJ has been reported to be most active for donation decisions that were associated with the maximum financial cost each individual was willing to pay for the transfer; the TPJ was less active for the same donations when they cost less and were thus accepted or when they cost more and were not accepted (*Morishima et al., 2012*). If TPJ activity thus serves to signal the conflict associated with the material cost/benefit of the donation, then TMS-related reductions of this conflict signal may lead subjects to require more conflicting information in order to switch away from the default response (giving to the worthy cause and not giving to the unworthy cause). In other words, subjects with TMS-related disruption of the TPJ should require a higher financial cost before not accepting donations to a good cause, or higher financial compensation before accepting donations to a morally bad cause. This behavioral prediction clearly differs from that of the first account, which would predict higher acceptance of transfers to both morally good or bad organizations. However, the predictions of the conflict account would be consistent with existing proposals that specific brain networks may bias conflict between moral values and material benefits, as documented for stimulation of prefrontal cortex during honesty (*Maréchal et al., 2017*) or fairness-related behaviors (*Knoch et al., 2006*).

The third and last account relates to the rTPJ function in the motivation to generate a positive reputation. The rTPJ is thought to build meta-representations when judging people's perspective of our own behaviour (*Saxe and Kanwisher, 2003*) and when inferring other peoples' intentions (*Young et al., 2007*). According to this perspective, the rTPJ would be necessary to represent the image of ourselves that we want to project to others (self-image motivation). The fact that humans are motivated by how others perceive them is demonstrated by the 'audience effect', a phenomenon in which prosocial behavior is increased when people know they are watched by others (*Ariely et al., 2009*; *Izuma, 2012*; *Izuma et al., 2011*; *Izuma et al., 2010*). If self-image motivation was the main function of the rTPJ during donation decisions, then TMS-related disruption of the rTPJ should reduce the impact of making public as compared to private choices on transfers, since participants should be less concerned about the impact their behavior has on the image projected to the observers in the public condition. This should hold similarly for transfers to the good or the bad cause.

To test these three competing predictions, we used continuous theta-burst stimulation (cTBS) to transiently reduce rTPJ activity (*Huang et al., 2005*) before participants completed the paradigm. We selected the rostral rTPJ activated during several moral actions, as evident in previous fMRI

studies on social cognition (*Feldmanhall et al., 2014*; *Hare et al., 2010*; *Morishima et al., 2012*; *Schurz et al., 2014*; *Young and Saxe, 2009*). In two different matched groups, we applied cTBS either over the rTPJ or the vertex (to control for unspecific TMS effects) before participants performed the task designed to differentiate the potential roles of the rTPJ in helping. Our results show that the TPJ signals the conflict between moral and material values during altruistic choices. These findings oppose the view that the rTPJ contributes to altruism either by representing a general motivation to help (independent of who is receiving help) or by the motivation to build a positive reputation.

## Results

### Behavioral experiment

In an initial pilot experiment, 20 participants rated the moral value of 10 different, real organizations operating in Switzerland. We selected the two organizations that were most consistently judged as having morally 'right' or 'wrong' objectives (Likert scale −7 - + 7; cut-off ±4). One organization was 'Bread for all' (www.brotfueralle.ch), which fights against starvation in underdeveloped countries. The other organization, called 'Lewas' (www.lewas.ch), intends to expand population use of weapons. Importantly, both groups did not differ in how they valued the organizations (see *Figure 1—figure supplement 1*). Moreover, the ratings show that participants were clearly motivated to financially support the good organization and to not support the bad organization (as also evident in their default responses at the lower right-hand corner of the heatmaps in *Figures 2* and *3* below).

With these organizations included in our task design, we tested the rTPJ contributions to moral decisions by comparing choice behavior across two groups. One of these received cTBS over the rTPJ and the other over the vertex, to control for unspecific side-effects of TMS. To eliminate potential learning of moral decisions, we kept the two cTBS groups independent. Because our experimental setting could in principle induce physical consequences such as fatigue or headache, we also

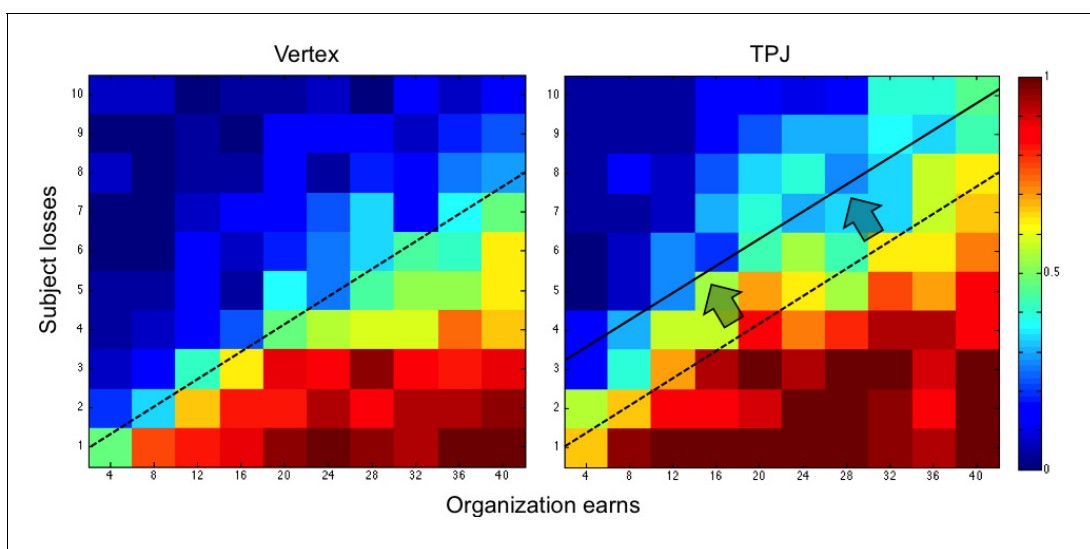

**Figure 2.** Color-coded map for probability of acceptance to donate (warm color shows greater acceptance probability; cold color lower acceptance probability) for the good organization in the control group (vertex) and TPJ group. Trials relative to both audience conditions (public/private) are shown. The black line represents the control group's responses at which donations were accepted with 50% probability for each given cost level. Vertical bar indicates the number of times a response was selected, that is one per subject.

DOI: https://doi.org/10.7554/eLife.40671.006

The following figure supplements are available for figure 2:

**Figure supplement 1.** Behaviour of individual subjects for the TPJ group and each organization type (good vs bad).

DOI: https://doi.org/10.7554/eLife.40671.007

**Figure supplement 2.** Behavior of individual subjects for the Vertex group and each organization type (good vs bad).

DOI: https://doi.org/10.7554/eLife.40671.008

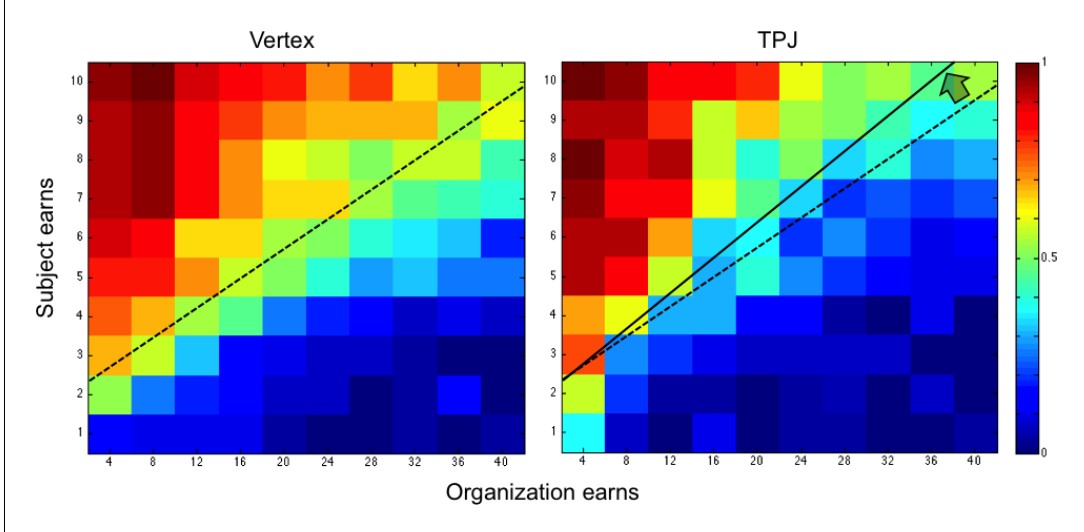

**Figure 3.** Color-coded map for probability of acceptance to donate (warm color shows greater acceptance probability; cold color lower acceptance probability) for the bad organization in the control group (vertex) and TPJ group. Trials relative to both audience conditions (public/private) are shown. The black line represents the control group's responses at which donations were accepted with 50% probability for each given cost level. Vertical bar indicates the number of times a response was selected, that is one per subject.

DOI: https://doi.org/10.7554/eLife.40671.009

evaluated mood changes from prior to after the experiment (Multidimensional Mood State Questionnaire, MMSQ). We found no significant differences between both cTBS groups (see **Supplementary file 1**).

## rTPJ stimulation leads to an increased impact of moral values.

**Figures 2** and **3** show the choice outcomes (proportion of accepted donations) for transfers to the good and the bad organization, respectively. These choice outcomes already show that participants by default were motivated to give to the good organization and to not give to the bad organization: when donations were large and associated with only minimal material consequences for participants themselves, all donations to the good organization were accepted and all donations to the bad organization were rejected (see lower right-hand corner of the heatmaps in **Figures 2** and **3**, respectively). Participants started to systematically switch away from this default response when transfers were coupled with larger material consequences for themselves, and these switching points were affected by the cTBS. To quantify these effects, we employed mixed-effect logistic regression to calculate how cTBS effects modulated participants' choices in the moral ('*good organization*') and immoral ('*bad organization*') contexts, depending on all choice parameters (the participants' monetary loss in the moral context or their monetary gain in the immoral context, the gain for the organization, the effect of cTBS, and the interaction between cTBS and the gain of the organization and cTBS). The model predicted the subject's probability *P(accept)* of accepting each specific offer during the donation task (**Table 1**) based on the offer's potential monetary value *Vs* (loss in the moral context or gain in the immoral context) to the subject and the gain *Go* for the organization on each trial. We set up separate models for choices concerning the good organization versus the bad organization, since these choices were paired in our design with diametrically opposite material consequences (material costs versus material gains, respectively). This required separate models with different regressors (resulting in parameter estimates of opposite valence). Our predictions took the following form (**Equation 1**):

$$P(accept) = \beta_0 + \beta_1 * V_s + \beta_2 * Go + \beta_3 * cTBS + \beta_4 * Go \, x \, cTBS \qquad (1)$$

where *Vs* is the monetary consequence of the transfer (material loss for good organizations or material gain for bad organizations), *Go* is the gain for the organization, and *cTBS* is a dummy variable indicating stimulation group. Employing this approach, we found that in the good moral

**Table 1.** Regression model results.
A, good organization; B, bad organization.

**(A)**

|  | Coefficient | Std. error | Z | P>|z| | 95% Conf. Interval | |
|---|---|---|---|---|---|---|
| Subject loss | −0.7989 | 0.0226 | 35.3 | **0.0001** | 0.7546 | 0.8433 |
| Organization gain | 0.1449 | 0.0120 | 12.04 | **0.0001** | 0.1213 | 0.1684 |
| cTBS | 1.3760 | 0.5994 | 2.3 | **0.022** | 0.2011 | 2.5509 |
| Gain organizationxcTBS | −0.0086 | 0.0075 | −1.14 | 0.254 | −0.0233 | 0.0061 |
| constant | 0.2288 | 0.4353 | 0.53 | 0.599 | −0.6244 | 1.0820 |

**(B)**

|  | Coefficient | Std. error | Z | P>|z| | 95% Conf. Interval | |
|---|---|---|---|---|---|---|
| Subject gain | 0.7415 | 0.0220 | 33.7 | **0.0001** | 0.6983 | 0.7846 |
| Organization gain | −0.2064 | 0.0133 | −15.43 | **0.0001** | −0.2327 | −0.1802 |
| cTBS | 0.0022 | 0.7063 | 0 | 0.997 | −1.3822 | 1.3866 |
| Gain organizationxcTBS | 0.0376 | 0.0079 | 4.72 | **0.0001** | 0.0220 | 0.0533 |
| constant | −1.6118 | 0.5126 | −3.14 | 0.002 | −2.6166 | −0.6069 |

Table 1(A):

**Equation 1**: logit (*prob. accept*) = $\beta o + \beta_1 * GS + \beta_2 * GO + \alpha_0 * cTBS + \alpha_1 * cTBS * GO$

Number of obs = 5800; Number of participants = 29; Obs per group: min = 200; max = 200; avg. = 200

Integration points = 7, Wald chi2(4)=1342.55

Log Likelihood = −1948.13, Prob > chi2=0.00001

Estimate: 1.65; Standard error: 0.21; cTBS: group

Table 1(B):

Number of obs = 5800; Number of participants = 29; Obs per group: min = 200; max = 200; avg. = 200

Integration points = 7, Wald chi2(4)=1294.59

Log Likelihood = −1902.89, Prob > chi2=0.00001

Estimate: 1.68; Standard error: 0.25; cTBS: group

DOI: https://doi.org/10.7554/eLife.40671.010

context, rTPJ stimulation increased the probability that participants accepted a given donation paired with a given monetary loss (cTBS Group coefficient = 1.37; p = 0.02; *Figure 2* and *Table 1A*). In other words, disruption of rTPJ functioning led participants to reduce monetary self-interest and to give away significantly more than the control group. In *Figure 2*, the black line represents the control group's indifference line at which donations were accepted with 50% probability for each given cost level; the significant change in the likelihood to accept is represented by a shift in this line between the TMS groups. Thus, this figure shows that for all possible donations levels, rTPJ stimulation lead participants to require a significantly higher cost to decide to switch away from their morally prescribed default to accept payments to the good organization.

In the bad moral context, rTPJ cTBS had a very different overall effect, since it significantly *reduced* the acceptance rate, particularly for high levels of gains to the organization (*Go* x *cTBS* coefficient = 0.03; p < 0.001; *Figure 3* and *Table 1B*). That is, participants in the cTBS group required significantly higher payments to switch away from the default of not accepting the transfer to the morally bad organization, as indicated by a slope change across the offer distribution (*Figure 3*). Again, this cTBS-induced change in choice behavior was expressed as a systematic shift of the choice indifference line (black line in *Figure 3*), but now in the inverse direction as compared to the good moral context (subjects required a higher monetary payoff to *accept* bad transfers versus subjects required a higher monetary cost to *reject* good transfers).

Overall, the cTBS effects on donation decisions are therefore inconsistent with the predictions of the 'other-regarding motivation' account, which would have predicted that cTBS leads to an overall decreased motivation to help and therefore lower levels of donations in both contexts. Instead, the pattern of results is consistent with the predictions of the 'conflict' account, since cTBS appeared to

have lowered the behavioral impact of the conflict between moral and material values, leading participants in both contexts to require a higher financial consequence (cost/benefit) to switch away from their morally motivated default choice (giving/not giving to the good/bad organization, respectively; cf. *Figures 2* and *3*).

## Reputation-based behavior is not affected by rTPJ stimulation

It is well established that prosocial behavior is increased when people know they are watched by others, which is often interpreted as reflecting reputation concerns (*Ariely et al., 2009*; *Izuma, 2012*; *Izuma et al., 2011*; *Izuma et al., 2010*). These reputation concerns may rely on the TPJ, in line with its presumed role in representing what others think our intentions are (*Saxe and Kanwisher, 2003*). However, although our data do reveal an audience effect presumably driven by reputation concerns, our analysis shows that this effect is not modulated by cTBS (*Figure 1—figure supplement 2*). We did not observe an interaction between cTBS and the audience condition, neither in the moral (cTBSxAudience coefficient = 0.04; p = 0.78; *Supplementary file 2A*) nor the immoral context (cTBSxAudience coefficient = −0.08; p = 0.59; *Supplementary file 2B*). In other words, participants in both groups reacted similarly to being watched while taking the moral decisions, making it unlikely that the stimulation changed the motivation to engage in reputation-enhancing behaviors. This provides evidence that the rTPJ's role in moral choices seems unrelated to the encoding of social reputation concerns.

Together, our results indicate that cTBS over the rTPJ increased the weight participants assigned to moral values over conflicting monetary values, irrespective of whether this conflict arises in situations with or without reputation concerns. This is consistent with the notion that the rTPJ signals the degree with which material values conflict with the morally prescribed default option (giving to a good cause and not giving to a bad cause), since a stimulation-related reduction in this signal would lead to the pattern of increased/decreased donations to the good/bad organization we observed here.

## Personality variables do not account for group differences

Traits, mood and social status are known to relate to an individual's propensity to help altruistically (*Guinote et al., 2015*; *Oda et al., 2014*; *Shaffer and Graziano, 1983*). To rule out that our results may have been biased by differences in the personality of subjects in the different groups, we compared the groups in terms of their personality characteristics, social value orientation, and their willingness to help others (see materials and methods). The experimental groups did not differ in these personality measures (*Supplementary file 1*). Moreover, we directly controlled for personality and questionnaires' influence in our statistical analysis, by repeating model 1 (*Equation 1*) with all personality covariates in *Supplementary file 1* added as regressors. This showed similar results to the original model, as the effects of group remain significant (moral context: cTBS coefficient = 1.31; p = 0.02; immoral context: cTBSxGo coefficient = 3.14; p < 0.001). Thus, cTBS results in both moral and immoral contexts cannot be attributed to (non-significant) group differences in prosocial attitudes, social interaction preferences, social value orientation or impulsivity.

## Discussion

Prosocial behavior and altruism may be governed by a general motivation to help others, by processes signaling the conflict between moral and material values, and by the motivational value of the reputation gained from helping. Here, we tested which of these three possible modulators of altruism may be causally instantiated by the rTPJ, a brain area commonly activated during altruistic acts. Our results contradict accounts positing that the rTPJ contributes to altruism via representing general other-regarding motivation to help (independent of the recipient's moral characteristics) or the motivation to build a positive reputation. Instead, they support the view that the rTPJ is involved in the representation of the conflict between moral and material values associated with a donation. This follows from the finding that cTBS of this brain area biased subjects to donate more readily to the good association but less readily to the bad association, suggesting that modulation of the rTPJ reduces the impact of financial motives that are in conflict with the default option prescribed by moral motives (helping worthy causes and not helping unworthy causes).

To the best of our knowledge, no prior study has provided direct causal evidence for a rTPJ role in representing the moral-material conflict associated with altruistic behaviour. However, prior neuro-imaging studies have already suggested that the rTPJ may serve as a neural hub that signals the conflict between self-interest and moral considerations. For instance, *Morishima et al. (2012)* showed that activity in this region was highest for donation decisions involving a monetary cost that just matched the participant's willingness to pay for this act, presumably entailing conflict between the motivation to help and the motivation to keep the money. Other social cognition studies of difficult moral decisions or choices involving others have shown increased activity in the rTPJ when decisions follow one's own moral values rather than other-oriented decisions (*Hare et al., 2010*; *Morishima et al., 2012*; *Park et al., 2017*; *Strombach et al., 2015*; *Yoder and Decety, 2014*). When subjects know who the recipient is, and the closer their social relationship to the recipient, the probability of helping increases and so does the TPJ activity (*Strombach et al., 2015*). Commitment to behave in a generous manner enhances TPJ activity during decisions to give away vs. gaining money to oneself (*Park et al., 2017*) and difficult moral decisions increase bilateral TPJ activity (*Feldmanhall et al., 2014*). Hence, the TPJ is active when decisions entail prosocial acts, but these studies do not establish a link between TPJ activity and the encoding of moral-material conflict.

Previous TMS studies have already pinpointed a causal role of the rTPJ in various social decisions (*Mengotti et al., 2017*; *Soutschek et al., 2016*; *Young et al., 2010*), but none of these studies makes quite the same point as our findings. That is, previous studies have suggested the rTPJ is responsible for modulating other-regarding preferences or beliefs in moral judgments, or that rTPJ stimulation results in enhanced cognitive abilities to generate thoughtful choices by overcoming egocentricity bias (using a perspective-taking and inter-personal/temporal tasks) (*Soutschek et al., 2016*). For example, one study assessed the rTPJ role in adjusting beliefs of right or wrong actions before and during decisions (*Mengotti et al., 2017*). Another study showed that TMS over the rTPJ resulted in subjects judging negative actions (e.g. harms) to be less morally harmful and more morally permissible (*Young et al., 2010*). While these studies thus reported changes in self-centeredness or general moral decision making, they did not investigate how the rTPJ's role in overcoming moral conflict when material and non-material values interact, as we did here.

While the results of previous studies appear generally consistent with our findings, the direction of our results (increase in moral behavior due to disruptive TMS) may at first glance seem counterintuitive. That is, most existing studies have attributed to the rTPJ functions such as processing other's perspective (*Hampton et al., 2008*; *Hill et al., 2017*; *Schurz et al., 2014*), conflict between self-centered and other-centered points of view (*Silani et al., 2013*), or processing of the moral deservingness of a recipient (*Hare et al., 2010*). One may think that disruption of these functions should result in decreased motivation to help others, or to follow moral values depending on the lower estimates of the recipient's moral deservingness. However, it is noteworthy that our study differs from these previous investigations in that the moral deservingness of the recipient was kept constant and clearly suggested a 'default' response (giving to the good organization and not giving to the bad organization) that we also observed in our choice data (see the unanimous acceptance of good transfers or rejection of bad transfers when these are coupled with minimal material consequences, lower right-hand corner of graphs in *Figures 2* and *3*). This implies that participants may have mainly paid attention to the varying financial cost/benefit and how it contrasts with the constant moral value of the donation: this may have predisposed the conflict signal in the rTPJ to mainly take into account this specific information. Future studies may investigate the impact of these methodological differences directly, by comparing accept/reject choices with fixed moral values as here with free choices based on varying moral values. In addition, it is important to highlight that the TMS effects on choices we observed here were expressed as a highly systematic shift of behavioral indifference lines toward higher material consequences of transfer/payoff pairs. This argues against possible alternative interpretations of our results in terms of other TMS effects unrelated to conflict, which would have made behavior noisier and less systematic. For instance, if TPJ TMS had affected number processing (*Cappelletti et al., 2014*), then it should have similarly affected processing of all magnitudes (both the transfers and the material consequences for the subject); this would not lead to a highly systematic shift of the indifference lines as observed here. The same logic argues against an explanation of our results in terms of TMS-induced distraction or task disengagement (*Kucyi et al., 2012*), which should have similarly reduced the systematic characteristics of the trade-off between the moral and material consequences of each transfer. Since this is not the case (compare the two panels in

*Figures 2* and *3*, standard errors of each model and no changes in raw RTs as observed in *Figure 1—figure supplement 3*), our results support the view that the rTPJ selectively mediates the impact of conflicts between the moral and the material consequences of a choice. This is a novel finding that may aid interpretation of many other moral functions linked to TPJ functioning.

The impact of the rTPJ conflict signal on behavior may not be mediated by local activity alone, but by directed communication with other brain areas. For instance, conflict-related modulation of the connectivity between rTPJ and the ventro-medial prefrontal (vmPFC) or ventral striatum may be responsible for modulating value computations, explaining increased helping decisions for the good cause (charity) and reduced monetary transfers to the bad cause. Indeed, functional connectivity between rTPJ, vmPFC and striatum is well documented (*Hare et al., 2010*; *Schurz et al., 2014*; *van den Bos et al., 2007*) and rTPJ activity disruption causes a change of its connectivity with vmPFC and other-related representations in the dorso-medial prefrontal cortex (*Hill et al., 2017*). To what degree similar functional interactions underlie the effects reported here could be examined by future studies using combined TMS and fMRI.

Independent of such considerations, we are in good position to rule out that the rTPJ contributes to altruistic choices by representing general other-regarding motivation to help or by facilitating social reputation-building. First, we provide direct and causal evidence that cTBS over the rTPJ did not modulate the general motivation to help as donations were increased only for the good organization but decreased for the bad organization. Second, rTPJ stimulation did not induce differential behavioral changes during the audience treatment (public vs private behavior). While the TPJ may play some role in reputation-based decision making (*Izuma, 2012*), our results show that this role does not encompass bringing to bear social reputation concerns during moral decisions. Perhaps, other regions that have been shown to be causally related to reputation concerns (such as the dorso-lateral prefrontal cortex; *Knoch et al., 2009*) are more plausible candidates for implementing this important aspect of behavior.

Resolving the moral conflict to help has been proposed to play an important function for the evolution of human prosocial behaviors (*Buckholtz and Marois, 2012*; *Trivers, 1971*). Functional coupling of mid-superior temporal sulcus and anterior cingulate gyrus increases with the size of social groups in monkeys (*Sallet et al., 2011*), perhaps paralleling the need for representation of one's role within a group and the adoption of the groups' moral values. Since stable communities enhance each individual's survival probabilities (*Trivers, 1971*), the TPJ's function may have developed to facilitate behavior that allows individuals to evaluate their self-interest while still remaining within the moral boundaries set by the groups' aims. Successful regulation of moral conflict may be key for maintaining group integration and thus survival (*Decety et al., 2004*; *Rilling et al., 2002*). Our results suggest that the brain has evolved mechanisms critical for dealing with the trade-off between self-interest and one's moral values, which may have been key during the development of modern societies.

Reduced consideration of others is present in autistic, psychopath and neurotic patients (*Harenski et al., 2009*; *Izuma et al., 2011*). Moreover, reduced activation in the rTPJ and the anterior temporal cortex has been reported in violent patients (*Wong et al., 1997*), pedophilia (*Massau et al., 2017*) and in psychopaths during moral decision-making tasks (*Harenski et al., 2009*); for review see *Fumagalli and Priori (2012)*. Thus, while the precise direction of TPJ stimulation effects on these behaviors still needs to be established, cTBS or other neuromodulation approaches could represent a potential therapeutical tool to induce a change in helping tendencies and prosocial behavior in some clinical populations.

To conclude, our study provides insights into the causal role of the rTPJ in contexts requiring management of moral conflict between personal and others interest. These insights refine our understanding of the rTPJ's functional contributions to prosocial behavior and may have important clinical implications.

## Materials and methods

### Participants

All data collection took place at the Laboratory for Social and Neural Systems research (SNS-Lab) at the University of Zurich, Switzerland. Thirty-two right-handed participants were included in the

experiment. All of them were fluent English speakers recruited from the University of Zurich database and gave informed written consent in accordance with local ethics, approved by the Kantonale Ethikkommission Zürich (2010-0326/3). Participants were included in the study following established TMS safety procedures (*Keel et al., 2001*). Data of subjects with fixed responses (accept/reject every response) was not included in the analysis due to biased preferential responses. One subject was excluded prior to the study due to recent migraine symptoms and two others were excluded after the experiment as they showed the fixed response pattern of rejecting all trials in the task.

## Experimental design

A between-subject design was used with 16 participants receiving cTBS over the rTPJ (age: 19–30 y, mean: 23.0; nine women; mean cTBS intensity: 38%) and 16 participants over the vertex area as a control target (age: 20–27 y, mean: 23.0; eight women; mean cTBS intensity: 40%). Subjects allocated to each stimulation group following strict random assignment. Power calculations based on two previous studies using cTBS over the rTPJ in human decision-making tasks (*Jeurissen et al., 2014*; *Young et al., 2010*) suggested a minimum sample size of 12 and 17 participants respectively per group to be required for an 80% probability of finding a significant effect (a = 0.05). Hence, our design included a total of 32 participants to satisfy statistical requirements.

To avoid potential habituation effects in donations during the charity task, we used a between-group design. Before the experiment, structural MRIs were obtained from all participants in a 3T scanner (181 sagittal slices, matrix size = 256 × 256, voxel size = 1 mm3, TR/TE/TI = 8.3/2.26/181 ms). Before the day of experiment, the participants completed an online survey with several questionnaires to evaluate prosocial and altruistic behaviors (see Supplementary material; *Supplementary file 1*). On the day of experiment, motor thresholds were obtained (see Supplementary material for details), followed by a mood questionnaire and organization ratings, task training, cTBS and task performance. The right TPJ was localized in each subject's MRI based on mean coordinates obtained from previous meta-analysis of functional MRI studies showing brain activation during several social psychology tasks (coordinates obtained from a pooled meta-analysis: social interaction [53 -31 9]; false belief [45 -59 39]; theory of mind [47 -36 9]; *Schurz et al., 2014*). Coordinates were individually adjusted for each subject to fully match anatomical landmarks (rostral of the caudal fissure and anterior to the angular gyrus). This resulted in the following average coordinates: [MNI: 48–42 19]. The vertex was manually localized in each participant's brain at the intersection between the midline and central sulcus. cTBS was administered with standard parameters (trains of 3 pulses at 50 Hz repeated every 200 ms; *Huang et al., 2005*) applied at 80% active motor thresholds from hand motor area (see Supplementary material for more TMS details). Immediately after the cTBS, participants performed the donation task, which lasted an average of 20.12 min (SD: 3.56, range 16–23) min.

## Donation task

Participants decided whether to accept or reject donations to two different organizations, either in presence or absence of an observer (*Figure 1A–B*). One organization was 'Bread for all' (www.brot-fueralle.ch), which fights against starvation in underdeveloped countries. The other organization, called 'Lewas' (www.lewas.ch), intends to expand the broader population use of weapons. Both are real organizations that operate in Switzerland; they were selected amongst 10 different Swiss organizations as being the most consistent in deservingness or non-deservingness to help (Likert scale −7 - +7; cut-off ±4) when tested in 20 participants from the same population. Participants received an initial endowment (100 CHF) that was modified after task completion: One public trial and one private trial were randomly selected at the end of task performance and the total payoff was modified (if trial was accepted) or not (if trial was rejected) by the corresponding monetary cost/benefit. The money going to the organization (if the trials were accepted) was sent by the experimenter and the participant received a written confirmation of the donation. Thus, participants knew their choices could let the experimenter send real money taken from the two random trials, which ensured incentive-compatibility of the choices.

Thus, on each trial, participants decided to accept or reject to contribute economically to the organizations. These contributions were always represented as gains to both organizations (from 4 to 40 Swiss Francs). However, these gains were coupled with a cost for the participants (from −1 to

−10 Swiss Francs) for choices for the good organization (*Figure 1C*) but a gain (from 1 to 10 Swiss Francs) when choices where proposed to the bad organization (*Figure 1D*). On half of the trials, participants took these choices while being watched by an observer (public condition), while on the other half, choices were not seen by the observer (private condition). The observer was in the same room and could watch the participant's choices on a second screen on public trials. The participants were explicitly informed about this fact during task practice. The task commenced with a block of donations to one of the two organizations (half of the participants started with the moral and half with the immoral organization), the ensuing blocks (25 trials per block) alternated between the good and bad organization. A total of 400 trials (200 per organization) were run, half of which were private and half public. *Figure 1C–D* shows all possible conditions and an example trial per condition as viewed by the participants.

During the experiment, subjects were presented several propositions to either donate to the positive organization or to the negative one (presented in blocks). The monetary amounts offered to the organizations and the costs/payoffs offered to subjects were varied independently across trials. Each trial offers for the organizations potential gains ranged from 4 to 40 francs, always in steps of 4 francs. Subjects potential costs/payoffs ranged from 1 to 10 francs, in steps of 1 franc. This manipulation resulted in 100 different dilemmas.

A key point of the task was to clearly induce in subjects the audience effect prior to task commencement. Prior to task practice, subjects were shown a secondary (photo if needed?) computer screen that was showing the task performance of the subject. Three reminders differentiated public from private trials to ensure subjects were fully aware of conditions. In public trials, a choice feedback was marked and highlighted, a cyan border surrounded the offer and human eyes were displayed above the offer (*Figure 1A*). Cues of being observed do influence participants' behavior (*Bateson et al., 2006*; *Haley and Fessler, 2005*). In private trials, however, no choice feedback was displayed, a yellow border surrounded the offer and a picture of a padlock was displayed on the top of the screen (*Figure 1B*). Each monetary dilemma was repeated once in public and once in private. Thus, subjects performed 400 decisions in total. 200 trials were for the positive organization and 200 for the negative one, while half of the times decisions were made in public, and half in private. Trials were block-ordered.

A trial event (*Figure 1A–B*) was presented as follows. Initially, an offer proposed to either accept or reject by pressing the left button response or the right button response on the computer keyboard, respectively. In public trials, the selected choice made was highlighted for 1 s by growing its characters, while the unselected choice disappeared. However, during private trials, no feedback was provided. Subjects were told to respond within 1 s, otherwise a message reminded them to respond. Instructions emphasized that accuracy was more important than speed during choices. Then, a fixation cross appeared during a random time interval, drawn from a uniform distribution between 1.5 and 3.5 s.

The task was presented in blocks counterbalancing by organization and audience conditions. The same organization was presented during two blocks (one in public, one in private in counterbalanced order), then switched to the other organization in the two following blocks (counterbalanced by public and private conditions). Thus, each block presented 25 dilemmas from the 100 possible combinations. The task was designed in Presentation software (Neurobehavioral Systems).

## Linear regression analysis

Separately for each organization, two models were fit to the data using mixed-effect regression analysis in STATA v.10. We set up separate models for choices concerning the good organization versus the bad organization, since these choices were paired in our design with diametrically opposite material consequences (material costs versus material gains, respectively). This required separate models with different regressors (resulting in parameter estimates of opposite valence). Each of the models predicted the probability of accepting an offer (logit), where model 1 (*Equation 1*, see results) tested the group's difference in acceptance rates for each context and model 2 (*Equation 2*) the cTBS effects on the audience effect.

The logit parameters in model 2 (*Equation 2*) test the likelihood of acceptance as influenced by the offer's potential monetary value where *Vs* is the monetary consequence of the transfer (material loss for good organizations or material gain for bad organizations), *Go* is the gain for the

organization, *cTBS* is a dummy variable indicating stimulation group, and *audience* is a dummy variable coding whether choices were observed or not.

$$P(accept) = \beta_0 + \beta_1 * V_s + \beta_2 * Go + \beta_3 * \text{Audience} + \beta_4 * \text{cTBS} + \beta_5 * \text{cTBS} \times \text{Audience} \quad (2)$$

Other models tested additional potential effects of cTBS on the decision variables. The full model took the following form and results presented in *Supplementary file 3*:

$$\begin{aligned} P(accept) &= \beta_0 + \beta_1 * V_s + \beta_2 * Go + \beta_3 * \text{Audience} + \beta_4 * \text{cTBS} + \beta_5 * \text{cTBS} \times \text{Audience} \\ &+ \beta_6 * cTBS \times V_s + \beta_7 * cTBS \times Go \end{aligned} \quad (3)$$

## TBS and MRI procedures

A Magstim super-rapid stimulator (Magstim, Whitland, Dyfed, UK) was used to stimulate both cortical sites using a 7 cm figure of eight-coil. To obtain each individual's TBS stimulation, the coil was placed tangentially over the subject's scalp with the handle pointing 45 degrees backwards and laterally over the right motor cortex. A quadricular grid was marked on each MRI using Brainsight over the hand motor region (located in the anterior portion of the central sulcus) to apply at least five pulses on each grid point and localize the most responding point to be selected as the hotspot. Responses to each grid point were measured by applying single pulses over the left hemisphere hand motor region and observing motor evoked potentials (MEPs) from the right first dorsal interosseous (FDI). The MEPs were triggered and continuously monitored by Brainsight amplifier with scalar feedback to ensure the pulses provoked clear. The stimulation coil was moved away from unresponsive grid points until a point showed consistent MEPs. Once the hotspot was selected, we asked subjects to activate their FDI by slightly pressing the thumb and index fingers to obtain their active motor threshold (AMT). The AMT was defined as the minimal stimulus intensity required to produce MEPs of $\geq 200$ mV amplitude in $\geq 5$ of 10 consecutive pulses.

After obtaining the stimulation output from the hotspot, a distance-correction for differences in scalp-cortex distance method was applied (*Stokes et al., 2007*). This protocol resulted in an average stimulation output of 39% (range 31–56), using intensity of 80% of each individual's AMT for the hand. Target sites were localized using the Brainsight frameless stereotaxy system (Rogue Research, Montreal Canada) with a Polaris (NorthernDigital, Waterloo, Canada) infrared tracking system. Within each subject's MRI, several anatomical landmarks were marked (bridge of nose, nose gap, left and right intra-trageal notches). The TBS coil handle had an infrared tracker to identify the scalp point where to orient coil center.

We used theta burst stimulation (TBS) applying repeated burst of 3 pulses at 50 Hz, known to deactivate the stimulated neurons during 30 min (*Huang et al., 2005*) when applied continuously. Every TBS burst was repeated at a 5 Hz rate resulting in 200 bursts with a total of 600 pulses. For TPJ TBS, the coil was placed with the handle perpendicular to the supramarginal gyrus anterior to TPJ (with the current going in the anterior to posterior direction) and for the vertex TBS it was held parallel to the midline. TBS was applied using online tracking system held by a robotic arm to guarantee correct placement during TBS period.

## Individual differences

Since the comparisons of interest were based between two groups of different subjects, we were aware of potential personality effects on our results. To check that the decisions were not influenced by individual differences in personality and predispositions for types of social behavior, we employed several questionnaires to measure the participants' personality characteristics (International Personality Item Pool, IPIP), Machiavellism levels (Mach IV Scale), social orientation (Social Value Orientation, SVO), impulsiveness (Barratt impulsive scale- III) and their willingness to help others (Prosocial Personality Battery, PSB). Personality characteristics that could interfere with social interactions (such as fear of being observed) were also assessed (Social Interaction Anxiety Scale, SIAS). Finally, we designed a questionnaire to obtain precise information of subject's beliefs about the experimental set-up (i.e. how observed they felt, how they perceived the cTBS, etc).

*Supplementary file 1* shows that groups had similar personality measures that did not differ significantly due to our randomization process. However, we directly tested the questionnaires influence in our statistical analysis in a repeated model 1 (*Equation 1*) but now adding all personality variables in *Supplementary file 1* as regressors. The results show similar results to the original

model where effects of group remain significant (moral context coefficient = 1.31; p = 0.02; amoral context TBSxGO coefficient = 3.14; p < 0.001). This confirms that the effects of TBS are real and significantly different between groups, and not due to any possible (non-significant) differences in personality measures.

### Rating of organizations

We obtained measures of each subject valuation of each organization to ensure no between-group differences coexisted in the behavioral results. Using a 7-likert scale, Familiarity, the monetary and personal implication, empathy and how subjects thought their friends will view the organizations. As expected, the good organization was rated higher overall than the bad organization (see *Figure 1—figure supplement 1*). Importantly, the valuation of organizations was not different between the cTBS and sham groups (F > 1).

## Acknowledgements

This work was financed in the framework of the Laboratory of excellence LABEX ANR-11-LABEX-0042 of Université de Lyon, within the program 'Investissements d'Avenir' (ANR-11-IDEX-0007) operated by the French National Research Agency (ANR) and this research has benefited from the financial support of IDEXLYON from Université de Lyon (project INDEPTH) within the Programme Investissements d'Avenir (ANR-16-IDEX-0005). This work was also supported by grants from the ANR-14-CE13-0006 BrainChoice, and the NSF-ANR CRCNS 'Social POMDP' (ANR-16-NEUC-0003-01) to J-C D. CCR received funding from the European Research Council (ERC) under the European Union's Horizon 2020 research and innovation programme (grant agreement No 725355, ERC consolidator grant BRAINCODES). We thank Pierre Wydoodt for help in analysis of the data, the recruitment team and technical staff at the SNS lab, University of Zurich, for their help with the conduction of the study.

## Additional information

### Funding

| Funder | Grant reference number | Author |
| --- | --- | --- |
| Horizon 2020 | 725355 | Christian, C. Ruff |
| Agence Nationale de la Recherche | 11-LABEX-0042 | Jean-Claude Dreher |
| Agence Nationale de la Recherche | 14-CE13-0006 | Jean-Claude Dreher |
| Agence Nationale de la Recherche | ANR-16-NEUC-0003-01 | Jean-Claude Dreher |
| Agence Nationale de la Recherche | ANR-16-IDEX-0005 | Jean-Claude Dreher |

The funders had no role in study design, data collection and interpretation, or the decision to submit the work for publication.

### Author contributions

Ignacio Obeso, Conceptualization, Software, Formal analysis, Investigation, Visualization, Methodology, Writing—original draft, Writing—review and editing; Marius Moisa, Data curation, Investigation, Methodology; Christian C Ruff, Conceptualization, Supervision, Investigation, Writing—review and editing; Jean-Claude Dreher, Conceptualization, Supervision, Funding acquisition, Validation, Writing—original draft, Project administration, Writing—review and editing

### Author ORCIDs

Ignacio Obeso (iD) https://orcid.org/0000-0001-8783-7281
Marius Moisa (iD) http://orcid.org/0000-0001-9789-3383

Christian C Ruff [iD] https://orcid.org/0000-0002-3964-2364
Jean-Claude Dreher [iD] http://orcid.org/0000-0002-2157-1529

## Ethics

Human subjects: Consent information was obtained prior to study commencement in every participant. The study was approved by the Research Ethics Committee of the canton of Zurich (2010-0326/3)

## Decision letter and Author response

Decision letter https://doi.org/10.7554/eLife.40671.018
Author response https://doi.org/10.7554/eLife.40671.019

## Additional files

### Supplementary files

• Supplementary file 1 Questionnaire scores for both experimental groups.
DOI: https://doi.org/10.7554/eLife.40671.011

• Supplementary file 2 Full regression audience model. A, good organization; B, bad organization.
DOI: https://doi.org/10.7554/eLife.40671.012

• Supplementary file 3 Full regression model. A, good organization; B, bad organization.
DOI: https://doi.org/10.7554/eLife.40671.013

• Transparent reporting form
DOI: https://doi.org/10.7554/eLife.40671.014

### Data availability

The data supporting the findings are available on Open Science Framework (https://osf.io/evh37/?=b2c6833f8b9e4622a2bd338e6c23c076).

The following dataset was generated:

| Author(s) | Year | Dataset title | Dataset URL | Database and Identifier |
|---|---|---|---|---|
| Ignacio Obeso, Marius Moisa, Christian Ruff, Jean-Claude Dreher | 2015 | Brain Stimulation to modify moral decisions | https://osf.io/evh37/ | Open Science Framework, evh37 |

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
