## [Decision Letter]

Thank you for submitting your article "A causal role for right temporo-parietal junction in signaling moral conflict" for consideration by *eLife*. Your article has been reviewed by three peer reviewers, including Christian Buchel as the Reviewing Editor and Reviewer #1, and the evaluation has been overseen by Michael Frank as the Senior Editor.

The reviewers have discussed the reviews with one another and the Reviewing Editor has drafted this decision to help you prepare a revised submission.

Summary:

This study investigated the role of the temporo-parietal junction (TPJ) in moral decision making. The three tested hypotheses were 1) motivation to help 2) representing conflicts between moral and material values and 3) representing social reputation concerns. Based on their data the authors conclude that rTPJ TMS interferes with a moral-material conflict, during altruistic choices, but not with a general motivation to help or by trying to look good in front of an audience.

This is a very interesting finding which sheds new light on the role of rTPJ in moral decision making. Using TMS as a virtual lesion tool with a clever behavioural task also allows to make statements on causality.

However, the reviewers identified a number of points that need to be addressed.

Essential revisions:

1) The task seems to trigger social reputation concerns only to a small degree (simple main effect in only one statistical model). Given that the authors rule out that social reputation concerns are related to rTPJ function it is important to show that the task indeed elicits social reputation concerns. This needs to be clarified.

2) The paper reports different statistical models, but it seems as if the model does not combine the good and the bad company in a single model. This would be required to formally test for interactions. Please provide a single statistical model or motivate using different models.

3) The reviewers were not convinced that the first hypothesis (helping motivation hypothesis) has been consistently linked with rTPJ function. This should be dropped or more evidence should be presented.

4) Their data statistically support the moral-material conflict hypothesis, but the results disagree with previous data which suggest that disruption of TPJ leads to maximizing material payoff at the expense of prosocial inclinations. This interpretation critically hinges on their assumption that TMS participants resorted to a suspected default response (giving to good charities, not giving to bad charities). However, because the authors did not measure what the participants' default response actually was, this needs clarification.

5) TPJ has also been implicated in numerical processing, day dreaming, default mode network and other processes, an alternative explanation of the TPJ-TMS effects on charitable giving is a reduction of the decision weight attached to monetary costs. This could be the consequence of a deficit in numerical processing, simply because TMS participants could not appropriately assess the numerical part of the cost information anymore. Likewise, resorting to the default response of giving to good causes and not giving to bad causes might also be the consequence of task disengagement. These alternatives should be discussed.

---

## [Author Response]

Essential revisions:1) The task seems to trigger social reputation concerns only to a small degree (simple main effect in only one statistical model). Given that the authors rule out that social reputation concerns are related to rTPJ function it is important to show that the task indeed elicits social reputation concerns. This needs to be clarified.

We agree with the reviewer that the reader needs to know to what degree our task elicits social reputation concerns. We now make this explicit by presenting a new figure (Figure 1—figure supplement 2) illustrating the data that support the presence of an audience effect (i.e., reputation concerns) (Izuma, 2012). The figure shows that the audience effect does not differ between the brain stimulation conditions, which supports the view that the TPJ does not seem to play a role in managing social reputation concerns. In addition to the figure, we have also added some new text:

“It is well established that prosocial behavior is increased when people know they are watched by others (Ariely et al., 2009; Izuma, 2012; Izuma et al., 2011, 2010). […] However, our analysis shows that the rTPJ’s role in the present donation decisions did not appear related to the management of reputation concerns (although an audience effect due to reputation concerns was visible: Public – Private donations, Figure 1—figure supplement 2).”

2) The paper reports different statistical models, but it seems as if the model does not combine the good and the bad company in a single model. This would be required to formally test for interactions. Please provide a single statistical model or motivate using different models.

The reviewer correctly notes that we analyze donations to the good and the bad organization with different models. This choice is intentional and required, since donations to the charity are paired with different outcomes for the participant in the two conditions: For bad charities, they are paired with monetary gains, whereas for good charities, they are paired with monetary losses. Thus, we cannot directly compare choices for good and bad organizations within one single choice model, since they are associated with different sets of regressors (costs in one condition versus benefits in the other). We appreciate that this may not have been clear from our initial descriptions and we have now added the following text:

“We set up separate models for choices concerning the good organization versus the bad organization, since these choices were paired in our design with diametrically opposite material consequences (material costs versus material gains, respectively). This required separate models with different regressors (resulting in parameter estimates of opposite valence). Our predictions took the following form (Equation 1):

P(accept) = βo + β_1_ * Vs + β_2_ * Go + β_3_ * cTBS + β_4_ * Go x cTBS (Equation 1)

where Vs is the monetary consequence of the transfer (material loss for good organizations or material gain for bad organizations), *Go* is the gain for the organization, and *cTBS* is a dummy variable indicating stimulation group.”

The following was added for Equation 2:

“where V_s_ is the monetary consequence of the transfer (material loss for good organizations or material gain for bad organizations), *Go* is the gain for the organization, *cTBS* is a dummy variable indicating stimulation group, and audience is a dummy variable coding whether choices were observed or not.”

3) The reviewers were not convinced that the first hypothesis (helping motivation hypothesis) has been consistently linked with rTPJ function. This should be dropped or more evidence should be presented.

We agree with the reviewer that the terminology used in the existing literature is often fuzzy, and that it can therefore be difficult to unambiguously assign different pieces of evidence to just one of the three hypotheses. Nevertheless, several studies have implicated the TPJ in “altruistic” or “other-regarding” behavior, suggesting that this structure underlies the motivation to give up material self-interest to help others. However, the concept of “altruism” is so widely defined that some readers confuse it with the motivation for “moral” behavior, as we have experienced when presenting these data. This generalization is problematic, since it may lead to diametrically different predictions regarding donations to the bad organization. In order to avoid these conceptual ambiguities, we employed different terminology to unambiguously focus on the aspect of altruism that lies at the heart of the proposed TPJ contribution to donation behavior (giving up self-interest for the benefit of others). With hindsight, we appreciate that this focus on “helping” may have appeared too narrow given the evidence in the existing studies. We therefore now employ the term “other-regarding motivation”, explain much more explicitly what the hypothesis refers to, and how it is backed up by existing proposals that include new studies:

**“**As for the first account, many recent findings are consistent with the interpretation that the rTPJ underlies other-regarding motivation during choices to do good to others. […] This account would therefore predict that TMS should bias participants to transfer less money to either ‘good’ or ‘bad’ recipients, reflecting a reduced general motivation to help.”

and:

“Overall, the cTBS effects on donation decisions are therefore inconsistent with the predictions of the “other-regarding motivation” account, which would have predicted that cTBS leads to an overall decreased motivation to help and therefore lower levels of donations in both contexts. Instead, the pattern of results is consistent with the predictions of the “conflict” account, since cTBS appeared to have lowered the behavioral impact of the conflict between moral and material values, leading participants in both contexts to require a higher financial consequence (cost/benefit) to switch away from their morally motivated default choice (giving/not giving to the good/bad organization, respectively; cf. Figures 2 and 3). “

4) Their data statistically support the moral-material conflict hypothesis, but the results disagree with previous data which suggest that disruption of TPJ leads to maximizing material payoff at the expense of prosocial inclinations. This interpretation critically hinges on their assumption that TMS participants resorted to a suspected default response (giving to good charities, not giving to bad charities). However, because the authors did not measure what the participants' default response actually was, this needs clarification.

As for the first part of the reviewer’s question (disagreement with existing findings), we are not aware of any existing study that directly shows that “disruption of TPJ leads to maximizing material payoff at the expense of prosocial inclinations”. Young et al. (Young et al., 2010) showed that TMS made subjects change their self-reported moral beliefs and Jeurissen et al., (Jeurissen et al., 2014) demonstrated that TMS affected moral-impersonal choices in hypothetical moral dilemmas. However, neither of these studies explicitly measured how TMS changes trade-offs of material self-interest versus moral motives (because there are no material consequences of reports or choices in these other studies). We now discuss this important difference between our approach and existing studies to avoid any misunderstandings:

“In addition, it is important to highlight that the TMS effects on choices we observed here were expressed as a highly systematic shift of behavioral indifference lines towards higher material consequences of transfer/payoff pairs. This argues against possible alternative interpretations of our results in terms of other TMS effects unrelated to conflict, which would have made behavior noisier and less systematic […]”

With respect to the second part of the reviewer’s question (measures of default responses), we agree that the manuscript should explicitly show the participants’ default response tendencies for the two charities. Our design did actually contain several measures of these default responses, but we may not have highlighted these enough to unambiguously make this point. We therefore now explain how the (opposite) default responses are measured by our design and can be read out from the lower right-hand corner of the choice matrices shown in Figures 2 and 3. That is, when there are only minimal material consequences, participants choose the default to always accept large donations to the good charity or to always reject large donations to the bad charity. Participants only switch away from these default choices when the material cost/benefit for the choices is large enough that it outweighs the moral considerations. Note also that the defaults revealed by choices under purely moral concerns fully concur with the value ratings for both organization types shown in Figure 1—figure supplement 1, which show that participants rated the good organization as much more worthy of support and as showing much more adequate activities. To avoid any ambiguities in this respect, we now explain these ratings in the legend of Figure 1—figure supplement 1 and we describe the default data and the underling logic in more detail as follows:

“Moreover, the ratings show that participants were clearly motivated to financially support the good organization and to not support the bad organization (as also evident in their default responses at the lower right-hand corner of the heatmaps in Figures 2 and 3 below).”

“Figures 2 and 3 show the choice outcomes (proportion of accepted donations) for transfers to the good and the bad organization, respectively. […] Participants started to systematically switch away from this default response when transfers were coupled with larger material consequences for themselves, and these switching points were affected by the cTBS.”

“that we also observed in our choice data (see the unanimous acceptance of good transfers or rejection of bad transfers when these are coupled with minimal material consequences, lower right-hand corner of graphs in Figures 2 and 3).”

Also added in the Results section:

“switch away from the default of not accepting the transfer to the […]. Again, this cTBSinduced change in choice behavior was expressed as a systematic shift of the choice indifference line (black line in Figure 3), but now in the inverse direction as compared to the good moral context (subjects required a higher monetary payoff to accept bad transfers versus subjects required a higher monetary cost to reject good transfers).”

5) TPJ has also been implicated in numerical processing, day dreaming, default mode network and other processes, an alternative explanation of the TPJ-TMS effects on charitable giving is a reduction of the decision weight attached to monetary costs. This could be the consequence of a deficit in numerical processing, simply because TMS participants could not appropriately assess the numerical part of the cost information anymore. Likewise, resorting to the default response of giving to good causes and not giving to bad causes might also be the consequence of task disengagement. These alternatives should be discussed.

We appreciate these suggestions for alternative ways to interpret our data, some of which we had also discussed internally during the write-up of our study. Given that the reviewers mentioned them, we discuss them explicitly in our Discussion. This new text makes it clearer why these alternative ways to interpret our data are unlikely to account for the systematic shift in the trade-off between the moral and material consequences of the donations. In short, we are not aware of any evidence that TPJ TMS changes numerical processing. Even if this were the case, any possible TMS effect on numerical processing would have equally affected the processing of both monetary amounts at stake (i.e., costs or benefits for both the subject and the charity). This can therefore not explain why the trade-off of these two quantities should shift in one direction. As for any possible effect of TPJ TMS on task disengagement, we also struggle to see how this could produce the systematic shift in trade-offs rather than more noisy / less consistent responses. Our data do not give any evidence that TMS made participants disengage from the task, as evidenced by random effects parameters of each model by means of standard error estimates (good organization: 0.21; bad organization: 0.25; details now included in each table for clarity and to provide the reader with exact values of our reasoning) and reaction times (see Figure 1—figure supplement 3). Instead, the systematic shifts in the tradeoff between material and moral consequences of the choice is more consistent with the notion that the TPJ signals the conflict between both option dimensions, as outlined in the paper. We discuss this reasoning in the new text:

“In addition, it is important to highlight that the TMS effects on choices we observed here were expressed as a highly systematic shift of behavioral indifference lines towards higher material consequences of transfer/payoff pairs. […] Since this is not the case (compare the two panels in Figures 2 and 3; standard errors of each model and no changes in raw RTs as observed in Figure 1—figure supplement 3),”